# OpenReview forum: "From Correction to Mastery: Reinforced Distillation of Large Language Model Agents"
_ICLR.cc/2026/Conference — Submitted to ICLR 2026_

### Official Review · Reviewer_WpXG · 2025-10-27

**Soundness:** 3
**Presentation:** 3
**Contribution:** 2
**Rating:** 4
**Confidence:** 3

**Summary:**

This paper introduces SCoRe (Student-Centered one-step Reinforcement), a novel framework for distilling the capabilities of large, powerful LLM agents into smaller, more efficient models. Unlike traditional Behavior Cloning, where a student passively imitates full teacher trajectories, SCoRe adopts a student-centric approach. It first initializes the student with basic skills and then enters a Mentored Problem-Solving phase. Here, the student attempts tasks independently, and the teacher only intervenes to correct the first error it makes. The student then continues from this corrected point. This method generates capability-matched training data that explicitly targets student weaknesses, theoretically reducing compounding error growth from O(H²) to O(H). A subsequent reinforcement learning phase further refines the model using short-horizon rollouts from verified prefixes and targeted key-step rewards, promoting independent problem-solving over mere imitation. Extensive experiments on mathematical, factual, and deep-search benchmarks show that small 7B-8B models trained with SCoRe can achieve performance comparable to a 72B teacher, significantly outperforming both standard distillation and other RL baselines.

**Strengths:**

1. The paper introduces a student-centered correction framework (SCoRe) that shifts from traditional teacher-driven imitation to a student-exploration–teacher-correction paradigm. This is a novel and conceptually elegant.
2. Experiments across 12 diverse benchmarks (reasoning, factual QA, deep research tasks) show that a 7B model distilled with SCoRe approaches the performance of a 72B teacher model.
3. The authors provide a clear theoretical analysis showing that SCoRe reduces error propagation from O(H^2) to O(H).

**Weaknesses:**

1. The framework's efficacy seems to be  contingent on a highly capable teacher model to provide accurate, minimal corrections. This creates a single point of failure; any errors or inconsistencies in the teacher's judgment are directly learned by the student.
2. The core "Mentored Problem-Solving" phase requires extensive, iterative interaction with a large teacher model (e.g., GPT-4) for data generation. This process is computationally expensive and incurs significant API costs, making the training process prohibitively costly despite the final student model being small.
3.  The method's cornerstone—correcting the "first" error—is often subjective in complex reasoning tasks. An initial flawed assumption may only manifest as a visible error steps later, making precise localization challenging. This ambiguity can lead to inconsistent training data and ineffective corrections.

**Questions:**

1. The entire SCoRe framework relies on a powerful teacher model. How would the performance degrade with a weaker or noisier teacher? Did you perform any sensitivity analysis on teacher quality?
2. Your method hinges on correcting the "first" error. In complex reasoning, an early logical misstep might only manifest as a visible error much later. How do you handle this inherent ambiguity, and did you observe cases where the "first error" was debatable?
3. How does SCoRe fundamentally differ from and improve upon classical interactive imitation learning algorithms like DAgger or HG-DAgger, which also aggregate student-generated data with expert labels?

---

> ### Author Response · Authors · 2025-11-16
> **Response to Reviewer WpXG (Part 1)**
>
> We sincerely thank the reviewers for their valuable feedback! Let's start with Weakness 3.
>
> ---
> ### **Weakness 3 and Question 2**:
>
> Your observation is absolutely correct! In our original paper, we addressed this issue with two key improvements:
>
> 1. **Global error localization**: Our method does not judge the "first error" midway through generation. Instead, the student finishes the whole trajectory first. Then the teacher reviews the entire reasoning chain and corrects the earliest deviation.  See Section 3.2 Student Explores, Teacher Corrects — especially the line “After the final answer, the teacher \pi_E checks correctness” — and Figure 2(a). This global analysis is specifically designed for situations where "An initial flawed assumption may only manifest as a visible error steps later", thereby avoiding subjective judgments based solely on local information.
>
> 2. **Corrected step verification via task success**: Once the teacher pinpoints the critical error and makes a single-step correction, the student keeps its original correct prefix and continues from the teacher's fix. If the task succeeds, this indirectly validates the teacher's localization: with the original prefix fully reserved and one teacher's correction step, the student still arrives at the correct answer.
>
> In the original paper (fourth paragraph of the Introduction and Section 3.2 Student Explores, Teacher Corrects), we described how errors are located globally and how the teacher's corrections are verified. In the rebuttal, with more space available, we updated both the Introduction and Methods to make this point clearer and emphasize it more strongly.
>
> ---
>
> ### **Weakness 1**:
>
> We apologize for the misunderstanding. Once Weakness 3 is clarified, this concern is also resolved.
>
> We do not always accept the teacher's corrections: **every correction must pass indirect verification.**
>
> Specifically, after the teacher provides a single‑step fix, the student resumes from its original correct prefix combined with that fix, and continues the task. The correction is considered valid only if the task is successfully completed; any fix failing this check can be removed from the training data.
>
> This safeguard ensures the student rarely learns from incorrect teacher judgments. Even if the teacher's fix is wrong, it is discarded during verification when the student still fails to complete the task after applying the fix.
>
> ---
>
> ### **Weakness 2**:
>
> After clarifying issues 1 and 3, note that the "iterative interaction" in MPS does not mean the teacher intervenes after every student step. The student first generates the entire trajectory, and the teacher interacts only afterward: keeping the actual interaction count low.
>
> As shown in Fig. 3, in MPS, only about 2.2% of training samples require two or more teacher interventions, already indicating low teacher‑call frequency.
>
> - Math tasks: one intervention: on average ~811 tokens/per data; two interventions: ~1275 tokens/per data.
> - Search tasks: one intervention: on average ~1260 tokens/per data; two interventions: ~2046 tokens/per data.
>
> Even at GPT‑4 API rates, these numbers are well within an acceptable cost. Moreover, in our experiments, we use Qwen2.5‑72B‑Instruct as the teacher, which can run locally on two high‑memory GPUs, eliminating expensive API calls altogether.
>
> ---
>
> ### **Question 3**:
>
> DAgger is an algorithm proposed in 2011, designed for traditional imitation learning tasks such as robot control. In contrast, SCoRe is designed for LLM agents that can autonomously call multiple tools. Because of the big differences in motivation, domain, and time of proposal, the two methods are almost entirely different.
>
> 1. DAgger is mainly an imitation learning method. It collects data and uses it for SFT. SCoRe uses data from the MPS phase in two very different phases: Phase 1: SFT based on corrections. Phase 2: Short-horizon RL. We start RL rollouts from the verified prefix before the first error. As shown in Theorem 3.3, this short-horizon strategy helps reduce gradient variance and improve training stability.
>
> 2. Even in reducing the compounding errors in Behavior Cloning, the methods are very different. In DAgger, the expert intervenes at every step when the student deviates, forcing the student back to the expert's path. In SCoRe's MPS phase, the student first generates a full trajectory alone. The teacher then reviews full trajectory and only corrects the first error. The student then continues with exploration from that corrected point.
>
> **Summary**: The two methods share only the high-level idea of reducing BC error accumulation. They differ greatly in task setting, data generation, training process, and theoretical properties.

---

> ### Author Response · Authors · 2025-11-16
> **Response to Reviewer WpXG (Part 2)**
>
> ### **Question 1**
>
> We would like to clarify that our method does not assume an unrealistically strong teacher. In fact, under certain conditions, **a less‑than‑perfect teacher can even benefit the RL phase**, because of the Hard‑to‑Teach data.
>
> *The brief definition of Hard‑to‑Teach data (Section 3.2 of the original or revised paper): In Mentored Problem-Solving (MPS), every teacher correction is automatically verified: after a one-step fix, the student resumes from its original correct prefix plus the fix. The correction is considered valid only if the task is successfully completed; If multiple teacher interventions still fail to produce a correct final answer, the example is labeled Hard‑to‑Teach.*
>
> In response to the reviewer's suggestion, we conducted an additional experiment on math tasks to examine how performance changes when the teacher model is less powerful: that is, when the proportion of Hard‑to‑Teach cases increases:
>
> | Hard‑to‑Teach ratio | AIME24 | AIME25 | MATH500 | OlymMath | Avg   |
> |---------------------|--------|--------|---------|----------|-------|
> | 0                 | 30.0   | 23.3   | 79.8    | 25.0     | 39.5  |
> | 0.1                | 36.7   | 26.7   | 82.0    | 26.5     | 43.0  |
> | 0.3               | 43.3   | 33.3   | 81.4    | 28.0     | 46.5  |
> | 0.5               | 30.0   | 23.3   | 81.4    | 27.0     | 40.4  |
>
> The results are surprising: a moderate increase (around 30%) in Hard-to-Teach data raised the average score, suggesting that challenging data can push the student beyond mere imitation of the teacher. This implies that even a less-than-perfect teacher can benefit the RL phase. However, if the teacher is too weak (e.g., Hard-to-Teach ratio above 50%), its ability to support the distillation process drops sharply, as such data lacks verified prefixes and key-step rewards for stability and is overly difficult. We consider such a weak teacher an extreme case in distillation.
>
> Thanks again for the valuable reminder!

---

> ### Author Response · Authors · 2025-11-26
>
> Dear Reviewer,
>
> We noticed that your earlier concerns came mainly from two misunderstandings: the teacher corrects the student after every single step, and teacher corrections are accepted without checking. In fact, our method locates errors only after **looking at the whole trajectory**, and we verify the teacher's correction by letting the student continue from its original correct prefix plus the teacher’s fix; the fix is kept only if the task can then be completed successfully. **Every correction must pass the verification**. We have also updated the paper accordingly to make these points clearer.
>
> We hope these clarifications help with your final review, and we would sincerely appreciate any further feedback when convenient.

---

> > ### Comment · Reviewer_WpXG · 2025-11-27
> >
> > thanks for the detailed response. and it address most of my concerns, and I would like to raise my score to 6.

---

### Official Review · Reviewer_tDd9 · 2025-10-30

**Soundness:** 2
**Presentation:** 2
**Contribution:** 2
**Rating:** 2
**Confidence:** 4

**Summary:**

Training a student model to imitate full teacher trajectories often suffers from compounding errors, since mistakes early in the rollout can accumulate. To address this challenge, this paper proposes SCoRe, a framework in which the student generates its own initial trajectory and the teacher intervenes at the earliest detected mistake, potentially iteratively. This enables more efficient transfer of agentic reasoning and tool-use capabilities from a large teacher LLM to a smaller student model. The authors claim this reduces compounding-error growth from $O(H^2)$ to $O(H)$ and improves RL stability by shortening rollout horizons. Empirical results across 12 benchmarks show that a 7B student model can match the agentic performance of a 72B teacher model.

**Strengths:**

1. The motivation to make LLM agents more affordable and deployable while preserving strong reasoning and tool-use capabilities is compelling.

2. The results clearly demonstrate consistent improvements over the base model across multiple benchmarks.

3. Figures and tables effectively illustrate the method and its empirical performance.

**Weaknesses:**

1. The proposed imitation-plus-correction paradigm resembles DAgger-style intervention in traditional RL literature and shares similarities with existing trajectory-repair approaches. Therefore, the degree of conceptual novelty may be limited without a more thorough discussion of prior work.

2. The method assumes a stronger teacher agent capable not only of solving the task but also of accurately identifying and correcting the student’s errors. This constitutes an even higher capability requirement than simply solving the task, and may limit applicability in scenarios where such a teacher is unavailable.

3. Finally, it is also not unclear how this method scales to frontier-level model improvement. Because the approach fundamentally relies on an even stronger teacher model, the path toward pushing beyond the teacher’s performance remains ambiguous.

4. The reported student performance appears noticeably weaker than existing smaller-scale models such as DeepScaleR-1.5B-Preview (43.1% on AIME2024, 87.8% on MATH500, 50.0% on OlympiaBench), which surpasses all student models up to 8B parameters in this paper. This raises concerns about the practical significance of the gains demonstrated here.

5. While the paper claims to minimize teacher involvement, the correction process is iterative, and the associated cost (latency, compute, prompts, dollar cost) is not quantified. It remains unclear whether the reduced rollout horizon translates into overall cost efficiency in real deployments.

6. Certain design choices lack justification or clarity. For example, the rationale behind setting the reward to 0.5 when the student’s key step matches the teacher’s correction is not explained, nor is the precise definition of a “correction” formalized.

**Questions:**

1. What's the Qwen3-8B-Instruct model referenced in Table 2?

2. Are there any failure modes on tasks where teacher guidance provides no improvement or even degrades performance?

3. How sensitive is the framework to formatting? In particular, how does SCoRe handle non-programmatic or free-form tasks, where determining the minimal teacher “leakage” needed to ensure effective learning may be challenging?

4. How does this approach compare with teacher-prefix training or mixed rollout BC+RL strategies that have been widely adopted in recent RL-finetuning work?

5. The theoretical analysis appears to evaluate expectations under the student’s on-policy distribution $d^{\hat{\pi}_i}$. However, since teacher actions are injected during rollout, isn’t the effective execution policy a mixture of student and teacher? Should the proof instead be derived under this mixture distribution?

---

> ### Author Response · Authors · 2025-11-16
> **Response to Reviewer tDd9 (Part 1)**
>
> We sincerely thank the reviewers for their valuable feedback. We have carefully addressed all comments and made the necessary revisions to the paper.
>
> ---
>
> ### **Weakness 1**:
>
> DAgger is an algorithm proposed in 2011, designed for traditional imitation learning tasks such as robot control. In contrast, SCoRe is designed for LLM agents that can autonomously call multiple tools. Because of the big differences in motivation, domain, and time of proposal, the two methods are almost entirely different.
>
> 1. DAgger is mainly an imitation learning method. It collects data and uses it for SFT. SCoRe uses data from the MPS phase in two very different phases: Phase 1: SFT based on corrections. Phase 2: Short-horizon RL. We start RL rollouts from the verified prefix before the first error. As shown in Theorem 3.3, this short-horizon strategy helps reduce gradient variance and improve training stability.
>
> 2. Even in reducing the compounding errors in Behavior Cloning, the methods are very different. In DAgger, the expert intervenes at every step when the student deviates, forcing the student back to the expert's path. In SCoRe's MPS phase, the student first generates a full trajectory alone. The teacher then reviews full trajectory and only corrects the first error. The student then continues with exploration from that corrected point.
>
> **Summary**: The two methods share only the high-level idea of reducing BC error accumulation. They differ greatly in task setting, data generation, training process, and theoretical properties.
>
> ---
>
> ### **Weakness 2**:
>
> We would like to clarify that our method does not assume an unrealistically strong teacher.
>
> -   Even if the teacher is weak, ****its incorrect fixes will be automatically removed during verification****. We verify the teacher's correction by letting the student continue from its original correct prefix plus the teacher’s fix; the fix is kept only if the task can then be completed successfully.  Every correction must pass verification, and incorrect fixes will not affect training.
>
> - In fact, under certain conditions, **a less‑than‑perfect teacher can even benefit the RL phase**, because of the Hard‑to‑Teach data.
>
> *The brief definition of Hard‑to‑Teach data (Section 3.2 of the original or revised paper): In Mentored Problem-Solving (MPS), every teacher correction is automatically verified; If multiple teacher interventions still fail to produce a correct final answer, the example is labeled **Hard‑to‑Teach**.*
>
> **"Hard‑to‑Teach" data benefit the RL training**: These Hard‑to‑Teach cases are not discarded. Part of them serves as high‑difficulty data in the RL phase, encouraging the student to explore solutions beyond the teacher's strategies. As shown in Fig. 3b of the paper, after RL training, the student's accuracy on validation subset of hard data can rise dramatically.
>
> - In response to the reviewer's suggestion, we conducted an additional experiment on math tasks to examine how performance changes when the teacher model is less powerful: that is, when the proportion of Hard‑to‑Teach cases increases:
>
> | Hard‑to‑Teach ratio | AIME24 | AIME25 | MATH500 | OlymMath | Avg   |
> |---------------------|--------|--------|---------|----------|-------|
> | 0                 | 30.0   | 23.3   | 79.8    | 25.0     | 39.5  |
> | 0.1                | 36.7   | 26.7   | 82.0    | 26.5     | 43.0  |
> | 0.3               | 43.3   | 33.3   | 81.4    | 28.0     | 46.5  |
> | 0.5               | 30.0   | 23.3   | 81.4    | 27.0     | 40.4  |
>
> The results are surprising: a moderate increase (around 30%) in Hard-to-Teach data raised the average score, suggesting that challenging data can push the student beyond mere imitation of the teacher. This implies that even a less-than-perfect teacher can benefit the RL phase. However, if the teacher is too weak (e.g., Hard-to-Teach ratio above 50%), its ability to support the distillation process drops sharply, as such data lacks verified prefixes and key-step rewards for stability and is overly difficult. We consider such a weak teacher an extreme case in distillation.
>
> Thanks again for the valuable reminder!

---

> ### Author Response · Authors · 2025-11-16
> **Response to Reviewer tDd9 (Part 2)**
>
> ### **Weakness 3**
>
> After clarifying Weakness 2, this concern is also addressed: **our approach does not rely on a perfect teacher**, and in fact, a less‑than‑perfect teacher can enrich the RL phase.
>
> - **Hard‑to‑Teach data accelerates student progress**: In the MPS stage, we retain Hard‑to‑Teach cases(examples where the student still fails to solve the task even after multiple teacher interventions), and use them in RL training. These challenging cases compel the student to search for solutions beyond the teacher's demonstrated strategies, often enabling capability gains beyond the teacher's performance.
>
> Notably, as shown in our earlier response to Weakness 2, training with 30% Hard‑to‑Teach data achieved higher math benchmark scores than in the original paper. As long as the teacher isn't too weak(In more 50% of the data, students' mistakes could not be corrected), it won't affect the distillation performance.
>
> - **Evidence of surpassing the teacher**: In several Deep Search benchmarks (e.g., GAIA and WebWalker), the SCoRe‑RL‑trained student actually exceeds the 72B teacher's performance. Table 2 excerpt:
>
> | Models | GAIA‑1 | GAIA‑2 | GAIA‑3 | GAIA‑Avg | HLE | xBench | WebWalker | Avg |
> |------|--------|--------|--------|----------|-----|--------|-----------|-----|
> | Qwen2.5‑72B‑Instruct（teacher） | 30.8 | 36.5 | 16.7 | 32.0 | 7.8 | 31.0 | 38.5 | 27.3 |
> | SCoRe‑RL（student） | 53.9 | 36.5 | 16.7 | 40.8 | 11.0 | 27.0 | 43.0 | 30.5 |
>
> ---
>
> ### **Weakness 4**
>
> Thank you for raising this valuable discussion. We would like to clarify:
>
> - In math tasks, our main student backbone is **Qwen2.5‑7B (Sept, 2024)** (also Qwen2.5‑3B, Llama 3.1‑8B), released in 2024. In contrast, DeepScaleR‑1.5B‑Preview and its base model DeepSeek‑R1‑Distill‑Qwen‑1.5B were released in 2025, built on a newer architecture and trained with stronger pre‑training and fine‑tuning data. A direct comparison is therefore not fair.
>
> *The reason we chose Qwen2.5‑7B instead of Qwen3-8B in math tasks:  Qwen2.5‑7B was certainly never exposed to AIME 2025 during training, making our evaluation more reliable.* For the newer deep search benchmarks (e.g., GAIA), we take the Qwen3‑8B as the base model.
>
> - Our math training set contains about 5k examples; DeepScaleR‑1.5B‑Preview used 40k math examples. This large gap in data volume also impacts achievable performance.
>
>
> - When we replaced the student model in SCoRe with the 2025 Qwen3‑8B, results on math benchmarks clearly surpassed DeepScaleR‑1.5B‑Preview:
>
> | Model               | AIME24 | AIME25 | MATH500 |
> |---------------------|--------|--------|---------|
> | SCoRe-RL (Qwen3‑8B)    | 63.3   | 46.7   | 89.2    |
>
> *Note: Our benchmark uses OlymMath[1], rather than Olympiad Bench. OlymMath is a newer, more challenging dataset.*
>
> *[1] Sun, H., Min, Y., Chen, Z., Zhao, W.X., Fang, L., Liu, Z., Wang, Z. and Wen, J.R., 2025. Challenging the Boundaries of Reasoning: An Olympiad-Level Math Benchmark for Large Language Models. arXiv preprint arXiv:2503.21380.*
>
> - One core baseline in our paper, ARPO, also fine‑tunes Qwen2.5‑7B. We use the results reported in the ARPO paper (under identical experimental settings), and SCoRe consistently outperforms them.
>
> ---
>
> ### **Weakness 5**
>
> As shown in Fig. 3, during MPS process, only about 2.2% of samples require two or more teacher interventions, indicating low teacher‑call frequency and thus low overall cost.
>
> To further address the reviewer's concern, we have added average token consumption per intervention (including both prompts and outputs):
>
> - Math tasks: one intervention: on average ~811 tokens/per data; two interventions: ~1275 tokens/per data.
>
> - Search tasks: one intervention: on average ~1260 tokens/per data; two interventions: ~2046 tokens/per data.
>
> In our experiments, the teacher model (Qwen2.5‑72B‑Instruct) is deployed locally, so no per‑token dollar cost is incurred. For reference, if using an API‑based commercial model such as GPT‑4, even in the most token‑intensive case, search tasks with two interventions, would cost ≤ 0.05 dollars per data.

---

> ### Author Response · Authors · 2025-11-16
> **Response to Reviewer tDd9 (Part 3)**
>
> ### **Weekness 6**
> Thank you for the reviewer's valuable comments.
>
> **Reward Scale**
>
> RL training is very costly, especially for search tasks where rollouts require heavy API calls. The common practice is to tune parameters on small reasoning tasks with low tool‑call cost, then apply them to full experiments.
>
> We ran a small test on 2k math samples to quickly check different reward scales. Results:
>
> | R_key Reward Scale| AIME24 | AIME25 | MATH500 | OlymMath | Avg   |
> |---------------------|--------|--------|---------|----------|-------|
> | 0.3                  | 30.0   | 23.3   | 80.0    | 23.5     | 39.2  |
> | 0.5                  | 33.3   | 23.3   | 79.8    | 24.0     |   40.1 |
> | 0.7                   | 30.0   | 20.0  | 78.2    | 23.0     |  37.8 |
>
> The reward scale at 0.5 gave the best average performance. We therefore use 0.5 in the paper.
>
> **The definition of "correction"**
>
> Section 3.2 Student Explores, Teacher Corrects contains the exact definition, and now we describe it in more detail:
>
> - The student first generates a full trajectory with its own policy but fails the task.
> - The teacher reviews the whole reasoning chain of the student, finds the first wrong step, and replaces it with the correct one.
> - The student keeps the original correct prefix and continues from this fixed step. If the final task is then solved, it confirms the teacher’s localisation was correct.
>
> In short, a "correction" means: the student could not solve the task at first; after one teacher's fix at a single step, the student immediately succeeded, showing that this corrected step was critical.
>
> ---
>
> ### **Question 1**
>
> The "Qwen3‑8B‑Instruct" in Table 2 refers to Qwen3‑8B (Hugging Face: ```Qwen/Qwen3‑8B```).  We added the "‑Instruct" suffix simply to keep naming consistent with models like Qwen2.5‑7B‑Instruct.
> Following the reviewer's suggestion, we have modified the corresponding description in the paper.
>
> ---
>
> ### **Question 2**
>
> Cases where teacher guidance provides no improvement do exist: these correspond to Hard‑to‑Teach tasks. We have verified that such tasks can benefit RL training in the above response.
>
> Performance degradation is unlikely. In the MPS process, if the teacher gives an incorrect single‑step correction, the student typically fails to complete the task with that guidance. Such data can be filtered out during MPS, and the guide of teachers is not used in subsequent training.
>
> ---
>
> ### **Question 3**
>
> Thanks for your valuable suggestion.  In fact, we have done free‑form tasks on deep search benchmarks.
> A hard example in GAIA benchmark:
>
> ```
> How many times was a Twitter/X post cited as a reference on the English Wikipedia pages for each day of August in the last June 2023 versions of the pages?
> ```
>
> This type of question requires the agent to:
> 1. Interpret a free‑form, natural‑language request.
> 2. Plan a multi‑hop search strategy (navigate Wikipedia revision history, locate citations, filter by date, and detect Twitter/X links).
> 3. Get and aggregate counts per day over an entire month
>
> Table 2 presents results on the deep‑search benchmarks (GAIA, HLE, XBench, WebWalker). Here are the excerpted results.
>
> | Method                       | GAIA‑1 | GAIA‑2 | GAIA‑3 | GAIA‑Avg | HLE  | XBench | WebWalker | Avg. |
> |------------------------------|--------|--------|--------|----------|------|--------|-----------|------|
> |...Part of the results...
> | ARPO                         | 53.9   | 32.7   | 16.7   | 38.8     | 8.8  | 25.0   | 30.5      | 25.8 |
> | SCoRe‑RL                 | 53.9   | 36.5   | 16.7   | 40.8 | 11.0 | 27.0   | 43.0      | 30.5 |
>
> ---
>
> ### **Question 4**
>
> In the original paper, we have already compared our method with mixed rollout BC+RL strategies(GRPO and ARPO, which are initialized by BC) in Table 1 and 2. Here are the excerpted results:
>
> | Method        | AIME24 | AIME25 | MATH500 | OlymM | HQA  | 2Wiki | MuSiQ | Bamb | Avg. |
> |---------------|--------|--------|---------|-------|------|-------|-------|------|------|
> |...Part of the results...
> | GRPO            | 23.3 | 26.7 | 78.0 | 25.0 | 59.0 | 76.1 | 30.6 | 68.4 | 48.4 |
> | ARPO            | 30.0 | 30.0 | 78.8 | 18.0 | 58.8 | 76.1 | 31.1 | 71.5 | 49.3 |
> | SCoRe‑RL        | 36.7 | 26.7 | 82.0 | 26.5 | 61.4 | 76.8 | 32.2 | 72.8 | 50.8 |
>
> For more experimental results, please refer to Tables 1 and 2 in the paper.
>
> ---
>
> ### **Question 5**
>
> Thank you for the reviewer's insightful observation.  In fact, this aspect has been considered in our analysis:
>
> - In Eq. (6), the misalignment is measured **per step**, rather than assuming all steps lie under the pure student distribution.
>
> - In SCoRe, the teacher intervenes only at a few steps with single‐step corrections, while the remaining steps are considered aligned. Consequently, when deriving the compounding‐error bound, we only need to evaluate the per‐step misalignment at those intervened steps, which is precisely reflected in the proof in Eqs. (18)–(20).

---

> ### Author Response · Authors · 2025-11-26
>
> Dear Reviewer,
>
> We hope you are doing well. We have provided clearer explanations and additional experiments addressing your main concerns. For example, our method does not rely on a "perfect teacher"; the teacher does not intervene at every step, but only after the student completes the full trajectory, and therefore the overall token cost remains low, etc. We have also updated the paper accordingly to make these points clearer.
>
>  We hope these clarifications help with your final assessment, and we would sincerely appreciate any further feedback at your convenience.

---

### Official Review · Reviewer_6vAV · 2025-10-31

**Soundness:** 3
**Presentation:** 3
**Contribution:** 3
**Rating:** 6
**Confidence:** 3

**Summary:**

This paper introduces SCoRe, a new distillation framework for LLM agents. Instead of the standard "teacher-acts, student-clones" paradigm, SCoRe places the student at the center of trajectory generation. The student explores tasks and the teacher corrects only the earliest error, producing "verified prefixes" and "key steps." In ScoRe, multiple post-training (SFT,RL) paradigms exists. The paper points out: on 12 challenging benchmarks, a 7B-parameter student distilled with SCoRe matches the agentic performance of a 72B-parameter teacher.

**Strengths:**

- Novel distillation paradigm: Different from traditional "Teacher Acts, Student Clones", this paper propose "Student Explores, Teacher Corrects". (Fig1)
- Comprehensive empirical results: Table 1,2 in paper demonstrate proposed method's effectiveness in mathematical reasoning, factual reasoning and deep search.
- Strong empirical results: The 7B student nearly matches a 72B teacher, demonstrating substantial efficiency gains.

**Weaknesses:**

- Lack of comparison with newer distillation methods: The paper mainly compares the proposed method with the traditional “Teacher Acts, Student Clones” paradigm.
- Distillation Paradigm: From this distillation paradigm, the student’s performance ceiling may be constrained by the teacher’s capability. I noticed a brief explanation of this issue in the introduction section. However, since that "the 7B student nearly matches a 72B teacher and even surpasses it on certain tasks," this explanation could be expanded and further deepened.
- Presentation: The full term of SCoRe should be given the first time it appears.

**Questions:**

1. Could the authors compare their method with the latest distillation approaches? If such a comparison is not necessary, could the authors provide an explanation for that?
2. Could the authors provide a more detailed and in-depth explanation for Weakness 2?

---

> ### Author Response · Authors · 2025-11-16
> **Response to Reviewer 6vAV**
>
> ### **Weakness 1 and Question 1**
>
> Thank you for suggesting a comparison with newer distillation methods. In Appendix C,  we have compared SCoRe‑RL with the latest **Agent Distillation[1]** in Table 5 of the original paper, Table 6 of the revised paper.
>
> - Same evaluation setup: identical math‑reasoning benchmarks (MATH500, GSM‑Hard, AIME, OlymMath), exact‑match metric, and model configs (Qwen2.5‑7B, Qwen2.5‑3B Instruct) as the original work. Note: We did not compare factual reasoning, as we use a Web Search API, generally stronger than the embedding‑based retrieval in Agent Distillation, and we wanted to ensure fairness.
>
> Consistent gains: SCoRe‑RL beats Agent Distillation on all benchmarks and both model sizes.
>
>     Qwen2.5‑7B: On average +7.5(49.3 vs 41.8).
>     Qwen2.5‑3B: On average +5.0（42.1 vs 37.1）
>
> - Our strong baselines are GRPO and ARPO RL algorithms, as multiple studies show RL is crucial for tool‑augmented agentic tasks, and typically far outperforms SFT‑based distillation.
>
> We appreciate the reviewer's feedback. Given the limited number of strong, publicly reproducible Agent Distillation baselines, we would be happy to add further comparisons if recent comparable methods have been proposed.
>
> *[1] Kang, M., Jeong, J., Lee, S., Cho, J. and Hwang, S.J., 2025. Distilling llm agent into small models with retrieval and code tools. arXiv preprint arXiv:2505.17612.*
>
> ---
>
> ### **Weakness 2 and Question 2**
>
> Thanks for your valuable question! We have indeed considered that a teacher's capability might limit the student's upper bound, and we try to address this by explicitly retaining Hard‑to‑Teach data for RL training.
>
> **Hard‑to‑Teach data in RL**
>
> Hard‑to‑Teach data are tasks that the student still fails to solve after several teacher corrections. We include these in RL training. Such cases push the student to attempt solutions beyond the teacher's strategies, enabling performance gains above the teacher's level. As shown in Figure 3b:
>
> - Original accuracy on Hard‑to‑Teach data: 0% (These are cases where the teacher fails to teach the student. Some are used for training, and others as validation samples).
> - After SCoRe‑RL: accuracy rose to 24.3%.
>
> Extra training on challenging problems is essentially a crucial path to surpassing the teacher.
>
> **Additional results on the Effect of increasing the proportion of Hard‑to‑Teach data**
>
> After reminder by the review,  we increased the proportion of Hard‑to‑Teach data and measured performance on math reasoning benchmarks:
>
> | Hard‑to‑Teach ratio | AIME24 | AIME25 | MATH500 | OlymMath | Avg   |
> |---------------------|--------|--------|---------|----------|-------|
> | 0                 | 30.0   | 23.3   | 79.8    | 25.0     | 39.5  |
> | 0.1       | 36.7   | 26.7   | 82.0    | 26.5     | 43.0  |
> | 0.3                 | 43.3   | 33.3   | 81.4    | 28.0     | 46.5  |
> | 0.5                 | 30.0   | 23.3   | 81.4    | 27.0     | 40.4  |
>
> Of course, an excessive proportion of Hard‑to‑Teach data can impair RL training, as such data lacks verified prefixes and key-step rewards for stability, and is overly difficult. However, a moderate increase (around 30%) raised the average score, indicating that challenging data can push the student beyond mere imitation of the teacher.
>
> Thanks for your valuable question again!
>
> ---
>
> ### **Weakness 3**
>
> Thanks for your reminder!  SCoRe (**S**tudent-**C**entered **o**ne-step **Re**inforcement). We have emphasized it in the revised paper.

---

### Official Review · Reviewer_BuNJ · 2025-11-01

**Soundness:** 4
**Presentation:** 4
**Contribution:** 4
**Rating:** 8
**Confidence:** 4

**Summary:**

The development of language models into language agents has motivated the exploration of smaller language models as agent backbones, in order to reduce latency and computational cost. This has led to the development of agent distillation, where structured trajectories from larger teacher models are distilled into smaller student models. However, the reasoning and knowledge gap between the student and teacher models often prevents the student from fully learning the teacher’s behavior, and even a single erroneous step can push the student into out-of-distribution states.

This work proposes SCoRe to overcome the limitations of agent distillation. In this framework, the student model generates reasoning trajectories while the teacher intervenes only to correct the earliest detected error, after which the student resumes reasoning. Since the trajectories are primarily generated by the student itself, this approach eliminates the capability and knowledge mismatch introduced in the conventional teacher-acts–student-clones paradigm. The work provides theoretical justification for the observed reduction in error and introduces methods to mitigate the limitations of reinforcement learning (RL) in training large language models.

**Strengths:**

The strengths of this paper are listed below:

1. The paper introduces a method to address the knowledge and reasoning gap between the student and teacher models in the agent distillation paradigm.

2. The proposed distillation pipeline not only utilizes the final correct trajectories but also leverages key-step correction preference pairs. These preference pairs provide dense rewards during RL training by offering feedback at earlier error points. This approach prevents the student model from being limited to pure imitation and instead encourages independent problem solving.

3. The authors provide a theoretical justification showing that SCoRe reduces the quadratic error growth observed in behavior cloning for long-horizon tasks to linear error growth.

4. They further provide theoretical justification that starting rollouts from verified prefixes reduces gradient variance, leading to a more stable and efficient RL optimization.

5. The authors evaluate their method on several mathematical and factual reasoning benchmarks and demonstrate strong performance gains. They also show results across different model families.

6. They conduct an extensive ablation study to justify the effectiveness of both short-horizon and key-step reward training in RL for mathematical and factual reasoning datasets.

**Weaknesses:**

The weaknesses of the paper are listed below:

1. Rewarding the model merely for avoiding its original mistake can be problematic for several reasons. First, it provides no guarantee of correctness; the new action may still be wrong but simply differ from the previous error. This can mislead the learning process, as the model is reinforced for being different rather than for being correct. Second, it introduces ambiguity in credit assignment, as the model might accumulate positive rewards for logically invalid steps. Overall, this design risks reinforcing superficial behavioral changes instead of genuine reasoning improvements, potentially leading to unstable or suboptimal learning dynamics.

2. It would be nice to check the performance of SCoRe on benchmarks where the model has not been trained on.

**Questions:**

1. Please clarify the use of R_avoid reward.
2. Please outline the limitations of your pipeline observed during the experiments.

---

> ### Author Response · Authors · 2025-11-16
> **Response to Reviewer BuNJ**
>
> ### **Weakness 1 and Question 1**
>
> Thanks for your valuable comments!
>
> - R_avoid is essentially **a format reward similar to that used in DeepSeek‑R1**, with a slight modification: R_avoid is granted only when the output is both format‑correct and different from the original error, and the reward is very small (0.1). In contrast, DeepSeek‑R1 assigns the reward(0.1) solely based on format correctness. This extra condition makes R_void stricter than DeepSeek‑R1.
>
> Because the magnitude of R_avoid is extremely small, and its primary role is simply to ensure format correctness, I think it has no adverse effect on training dynamics, just like DeepSeek‑R1.
> To prevent misunderstanding, we have explicitly stated the numerical value of R_avoid in the paper, and emphasize that it is merely a minor modification of the standard format reward in DeepSeek‑R1.
>
> - We appreciate the reviewer's insightful suggestion. We agree that both the format reward in DeepSeek‑R1 and our R_avoid are relatively simple. It is a challenging problem to design process‑level rewards for trajectories that ultimately fail to solve the task, but contain promising reasoning steps. We believe it is worth further exploration in future work.
>
> ---
>
> ### **Weakness 2**
>
> Thank you for suggesting OOD evaluation. We also care deeply about generalization.
>
> In fact, our experiments already include a task distribution completely absent from training data: **Deep Search** (GAIA, HLE, xBench, WebWalker).
>
> Our training set only has math/factual reasoning data, while Deep Search data differs sharply:
>
> - **Tool‑call frequency**: math/factual tasks average ~2.1 tool calls; Deep Search (e.g., GAIA) reaches ~4.8 or more.
> - **Tool‑type complexity**: Deep Search often requires web search + code in combination, versus single‑tool use in math/factual reasoning.
>
> | Method         | GAIA-1 | GAIA-2 | GAIA-3 | GAIA-Avg | HLE  | XBench | WebWalker | Avg  |
> |----------------|--------|--------|--------|----------|------|--------|-----------|------|
> | Behavior Cloning | 33.3   | 26.9   | 8.3    | 27.2     | 8.4  | 21.0   | 34.5      | 22.8 |
> | GRPO           | 48.7   | 25.0   | 8.3    | 32.0     | 7.8  | 20.0   | 29.0      | 22.2 |
> | SCoRe-RL       | 53.9   | 36.5   | 16.7   | 40.8     | 11.0 | 27.0   | 43.0      | 30.5 |
>
> **Performance (Table 2 in the paper):**
>
> - With Qwen3‑8B as the student, SCoRe‑RL clearly outperforms BC and GRPO, and even surpasses the 72B teacher's TIR prompting (30.5 vs 27.3).
> - On GAIA‑Avg, BC scores only 27.2, while SCoRe‑RL hits 40.8.
>
> If the reviewer would like to see results on other specific benchmarks, we are happy to provide them.
>
> ---
>
> ### **Question 2**
>
> Thanks for your valuable comments!
>
> Our main observation is that hard data is critical for effective RL training.
>
> During the MPS process, some samples remained unsolved by the student model even after multiple teacher interventions. Instead of discarding these cases, we deliberately retained them as challenging examples for RL, keeping their proportion at around 10%. As shown in Fig. 3, the experiments demonstrate that after RL training, the model can correctly answer previously unsolvable hard samples (which had been excluded from the training data). This suggests that exposure to hard data can enable the student model to surpass the teacher.
>
> However, collecting high‑quality hard samples is difficult. As shown in Fig. 3a, during the MPS process, only 11%–13.7% of the data are classified as hard samples.  One reason is that existing real‑world data may have already been seen by the model during pre‑training. Therefore, creating new and diverse hard samples is one of the limitations we identified in our experimental process, and we take it as an important area for future work.

---

> ### Comment · Reviewer_BuNJ · 2025-11-27
>
> Thank you for addressing most of my concerns. I will maintain my score.

---

### Meta-Review · Area_Chair_Vxz9 · 2026-01-09

**Summary:**

The paper studies the problem of compounding error in student-teacher distillation with full imitation of the teacher’s trajectories. In the proposed framework, called SCoRe, the student, initialized its capacity with cold-start behavior cloning and then supervised fine-tuned with high-quality teacher strategies, first generates its own trajectories, then the teacher identifies and corrects at the earliest error. These corrected trajectories are used for the next round of SFT. The paper shows that this reduces compounding-error growth from  $O(H^2)$ to $O(H)$ and achieves favorable performance across 12 benchmarks where a 7B student model can match the performance of a 72B teacher model.

The reviewers agree that the proposed approach is interesting and theoretically justified, with strong empirical results. The main concerns involve (1) similarity to DAgger-style intervention in RL, (2) correctness and consequences of assumptions around the teacher’s ability to correct errors, (3) additional experiments (e.g., evaluation on OOD data, computational cost, comparison with distillation methods, applicability to frontier models). The authors have made ean xtensive effort to address these concerns, and most of them have been sufficiently clarified. Nevertheless, I believe that the discussion on the related imitation work in RL and the analysis of the student’s better performance compared to its larger teacher is not adequately analyzed. In specific, the paper shares a similar high-level idea to DAgger; while they are in different domains and the authors briefly discuss their differences in the rebuttal, it’s unclear whether a trivial adaptation of similar imitation learning to the studied domain can be done. Furthermore, the explanation for “hard-to-teach” data’s benefit in pushing the student’s performance matching or even surpassing its larger teacher is not clearly described; as all learning signals are distilled from the teacher, what exactly makes the student achieve better capacity — this part is important but currently not clear from the paper. Consequently, I think the paper is not yet ready for publication and hope the authors will expand these important dimensions further in the next revision.

**Reviewer Concerns:**

- BuNJ
   - Rewarding the model merely for avoiding its original mistake can be problematic for several reasons.
       - First, it provides no guarantee of correctness;
       - Second, it introduces ambiguity in credit assignment, as the model might accumulate positive rewards for logically invalid steps
       -  this design risks reinforcing superficial behavioral changes instead of genuine reasoning improvements
   - It would be nice to check the performance of SCoRe on benchmarks where the model has not been trained on.
   - imitations of your pipeline observed during the experiments.
- 6vAV
   - Lack of comparison with newer distillation methods
   - From this distillation paradigm, the student’s performance ceiling may be constrained by the teacher’s capability, so explanation on “"the 7B student nearly matches a 72B teacher and even surpasses it on certain tasks” should be expanded.
- tDd9
   - proposed imitation-plus-correction paradigm resembles DAgger-style intervention in traditional RL literature -> limited novelty without proper discussion with related works
   - has even higher capability requirement than simply solving the task
       - `the teacher model is less powerful: that is, when the proportion of Hard‑to‑Teach cases increases` -> not sure if this is a good proxy for less powerful teacher.
   - unclear how this method scales to frontier-level model improvement.
   - The reported student performance appears noticeably weaker than existing smaller-scale models such as DeepScaleR-1.5B-Preview —> concern for practical significance?
   - the associated cost (latency, compute, prompts, dollar cost) is not quantified
   - the rationale behind setting the reward to 0.5 when the student’s key step matches the teacher’s correction is not explained, nor is the precise definition of a “correction” formalized.
   - any failure modes on tasks where teacher guidance provides no improvement or even degrades performance?
   - sensitivity to formatting?
   - comparison to teacher-prefix training or mixed rollout BC+RL strategies that have been widely adopted in recent RL-finetuning work
   - isn’t the effective execution policy a mixture of student and teacher?
- WpXG
   - The framework's efficacy seems to be contingent on a highly capable teacher model to provide accurate, minimal corrections.
   - extensive, iterative interaction with a large teacher model (e.g., GPT-4) for data generation
   - correcting the "first" error—is often subjective in complex reasoning tasks

Most of the concerns above are addressed. However, the response to the similarity to imitation learning and student's performance above the upperbound teacher's capacity are not fully explained.

**Reviewer Scores:**

- BuNJ - 8 (reasonable but limited concerns, due to positive score). Reviewer maintain score after authors’ responses.
- 6vAV - 6 (good comments, especially on why student model outperforms its teacher). Reviewer doesn’t respond.
- tDd9 - 2  (good comments). Reviewer doesn’t respond.
- WpXG - 4. Reviewer doesn’t respond.

Reviewers 6vAV would likely keep the score, while tDd9 woud likely increase score slightly at best (2->4). Reviewer WpXG would also not likely to increase the score.

---

### Decision · Program_Chairs · 2026-01-26

Reject